# The Selection of Leveler Parameters Using FEM Simulation

**DOI:** 10.3390/ma17010052

**Published:** 2023-12-22

**Authors:** Sebastian Mróz, Piotr Szota, Tomasz Garstka, Grzegorz Stradomski, Jakub Gróbarczyk, Radosław Gryczkowski

**Affiliations:** 1Faculty of Production Engineering and Materials Technology, Czestochowa University of Technology, 42-201 Częstochowa, Poland; piotr.szota@pcz.pl (P.S.); tomasz.garstka@pcz.pl (T.G.); grzegorz.stradomski@pcz.pl (G.S.); 2Serwistal Sp. z. o.o., 2A Dojazdowa str., 19-300 Ełk, Poland; j.grobarczyk@serwistal.pl (J.G.); r.gryczkowski@serwistal.pl (R.G.)

**Keywords:** leveling, flatness, numerical modeling, FEM, 3D scanning

## Abstract

The aim of this research was to select parameters for the roll pre-leveler to provide sheet metal waviness reduction after unwinding from the coil. Straightening parameters were selected based on the results of numerical simulations with the use of an FEM-based computer program. The material used for research was a hot-rolled sheet metal of grade S235JR + AR with a thickness of 3 mm and width of 1500 mm after unwinding from the coil. A mathematical model was developed to determine straightening roll arrangements in the pre-leveler. It enabled roll arrangement selection and a straightening scheme to be elaborated. The model’s innovative feature was conducting straightening numerical simulations for the real sheet metal geometric models obtained as a result of 3D laser scanning, which increased the accuracy of the numerical calculations.

## 1. Introduction

The sheets, after being unwound from the coil, are characterized by waviness and non-uniform residual stress levels in the cross-section and length thereof. This is, among others, the residue after the rolling process and winding of the sheet metal on the coil, so-called “coil memory”, which contributes to geometric defect formation and the emergence of elevated levels and residual stress non-uniform distribution [1,2].

The main cause of emerging uncontrolled sheet metal strains, e.g., after the cutting process, is residual stress non-uniform distribution [3,4]. Residual stresses are those that remain in the product after the removal of all loads, and they are introduced as a result of various technological operations, such as, for example, metal forming, heat treatment, casting, or welding; they also arise in machine and equipment components as result of operation thereof. Information on residual stresses was published for the first time in 1911 in a paper by Heyn and Bauer [5]. As a result, the uniaxial longitudinal residual stress condition in steel bars was determined. This marked the start of the scientific discipline dealing with influence analysis, formation origin, and residual stress measurement. In industrial practice, I-type stresses are of particular importance, i.e., stresses that can cover the entire research object. In the case of the residual stress level of sheet metal itself, first of all, anisotropy and principal direction orientation are essential in determining the suitability of the sheet metal for further processing [6]. The release of accumulated residual stresses in the sheet metal, for example, during longitudinal strip slitting processes or laser cutting processes, causes flexure of the cut-out elements, which may consequently lead to laser head damage [7,8].

Therefore, after unwinding, the sheet metal from the coil should be subjected to technological processes in order to, on the one hand, remove geometrical defects (flexure, edge, and central waviness, sickle-shaping form and twist-shaping form) and, on the other hand, to homogenize and minimize the residual stress level. The leveling process is a process that can eliminate the mentioned sheet metal defects [9]. Among the known technologies for straightening and residual stress level homogenization, various methods using roll levelers [10,11], tensioning [12,13,14], and skin-pass rolling [15,16] are widely used. The mentioned methods, besides ensuring sheet metal flatness, must also ensure the appropriate value of the permanent strain causing material transition into a yielded state. Material yielding as a result of setting the permanent yielding strain is very important, as these are strains that cause the initial stress state of the material obtained during the rolling process to be homogenized and directed. Yielding of the material also causes residual stresses to be relaxed [17,18]. Production of a sheet metal of the highest flatness requires defects after unwinding from the coil to be identified therein.

Flatness evaluation of the sheet metal dedicated for pre-straightening in the roll pre-levelers after unwinding from the coil is an important aspect because, in this stage of straightening, it is important to remove the substantial number of geometrical defects and to reduce the residual stress level. Introducing improvements related to the straightening process requires determining the sheet metal’s initial geometry. Most characteristic defects occurring in the cold- and hot-rolled sheets include a longitudinal bow—a type of curvature in the longitudinal direction, i.e., a crossbow—a type of curvature in the transverse direction, edge waviness that occurs when edges are longer than the strip center, and a sickle-shaping form and a twist-shaping form [19]. The above-mentioned defects may be eliminated to a great extent or completely by the roll straightening process in combination with straightening by tensioning or skin-pass rolling. However, determining the type of defect and the size thereof after unwinding from the coil is required in order to undertake an appropriate straightening method (in the case of the roll leveler, the positioning of the rolls) [20,21]. One cause of defect formation is irregular metal sheet rolling conditions and, in the case of hot-rolled sheet metal, an additional element influencing the formation thereof is temperature irregularity, which translates into different rolling conditions and then into sheet metal cooling. As a result of creating a non-uniform temperature distribution in the sheet metal, residual stresses are created which very often also influence the sheet metal geometry [10,17]. 

Using numerical modeling, the analysis of roller leveling is possible. In [22], the authors used FE analysis to depict the process of roller leveling, wherein it was possible to determine several combinations of roll positions, which resulted in a flat strip after leveling but showed different residual stress distributions across the strip thickness. The authors in [23] established a coupled curvature integration model that combines geometric coupling and curvature coupling. The variation in the curvature, the surface profile curve, and the plastic deformation rate of the sheet in a leveler were analyzed using the model. Two main conclusions were drawn: there are infinite combinations of entry intermesh values and exit intermesh values that make the residual curvature zero, and the variation in the entry intermesh value has a significant effect on the sensitivity of the residual curvature relative to the exit intermesh value. The authors in [24] present the modeling of multi-roller leveling for metal strips. To analyze the friction between the roll and the strip, the deformation curve of the strip was fitted, and the deformation function was constructed. With the help of the multi-roller leveler, the developed system was obtained more conveniently, and the related performance parameters were calculated quickly, which helped to introduce the digitalization of the complex multi-roller leveling technology. Perzynski et al. [25] used a computer-aided technology design using FE modeling and inverse analysis to evaluate the capabilities of roll levelers when applied to the leveling process of high-strength steel sheets. As a result of numerical modeling, the leveler setups were selected to match typical industrially available equipment. Also, in [26], a numerical method for eliminating residual stress by multi-roll leveling based on curvature coupling was discussed. The obtained results show that multi-roller leveling technology will cause rolling residual stress while reducing the initial residual stress of the sheet, and the larger plastic deformation caused by the intermesh of the work rollers at the entry is beneficial for the complete elimination of the initial residual stress, but the rolling residual stress will increase at the same time. Therefore, the total residual stress of the sheet after leveling depends on the appropriate leveling parameters. In [27], a model based on the curvature integration method was applied to an online plate leveling system. The developed quasi-static plate-leveling model can analyze the dynamic straightening process using a curvature integration method. The model can also analyze the straightening process of a plate with random curvature distribution. The authors in [28] proposed a new numerical model for a two-dimensional roller leveler which calculates the curvature and moment of the material depending on the intermesh. This curvature of plates was used to calculate the stress and strain values of the material along the thickness direction. The results from the developed numerical model and FE analysis were compared to verify the effectiveness of the model. The varying curvature ratio from FE analysis and the developed model according to the change in the maximum plastic strain for rolls were obtained. Gribkov et al. [29] developed an algorithm for a mathematical model that allows for the determination of the technological parameters of levelers. As a result of the calculation of the effect of the working rollers setup on the sheet metal roll’s quality during leveling on a multi-roll leveling machine, the laws for the rational rolls’ positioning were established. In summary, the use of modeling ensures the development of leveling technology. Thus, an engineering method and FE analysis were also used in the current paper to develop the leveling parameters.

Therefore, the aim of this research was the selection of roll pre-leveler parameters to ensure the removal of so-called “coil memory”, reducing waviness (geometry defects). Geometry identification of the sheet metal selected for research was carried out using an optical laser scanner, enabling waviness mapping of the selected sheet metal portion after unwinding from the coil to be elaborated. The sheet metal geometry measurements carried out using a 3D scanner allowed for the development of the sheet metal geometric models, which were then utilized for straightening process numerical modeling on the five-roller pre-leveler.

## 2. Materials and Methods

A hot-rolled steel of S235JR + AR grade was used for this research, as it is one of the most often used for structural components. Hot-rolled sheets of the S235JR grade can be annealed after the rolling process; however, due to the reduction in production costs, they are rolled into coils without additional heat treatment. During hot rolling, the rolls are cooled with water, and most often the edges of the sheet are cooled faster than the central part of the sheet, which may cause different properties in the central and side parts. The chemical composition of the steel is given in Table 1 [30].

A metal sheet with a thickness of 3 mm and width of 1500 mm was delivered in the form of coils. To determine yielding stress curves and Young’s modulus, static tensile tests were carried out. The MTS E45.305 (MTS Systems, Eden Prairie, MN, USA) machine with max. force of 30 kN was used. Static tensile tests were carried out with the use of the paddle specimens following applicable standards determining the performance of such tests [31].

The samples for mechanical properties testing were taken from sheets from the beginning, middle, and end of the coil in the middle sheet metal portion and near the sheet metal’s side edges. Three samples were prepared from each location. Due to the significant number of test samples, the testing methodology of two samples per area was assumed. In case of repeating results, the test series consisted of 2 samples. If a difference in tensile force values of more than 3% was obtained, a series including additional tests was used with the use of a third sample. A graphical representation of yielding stress curves is shown in Figure 1, whereas in Table 2 the mechanical properties of the steel grade tested are shown.

For the strain–stress curves obtained, exponential function form (1) with coefficients that most accurately represent real curves was assumed. Function coefficients of stress determination were chosen on the basis of approximation carried out with function (1) minimizing the square error. As in numerical modeling, the elasto-plastic model was used, i.e., during function coefficient determination, the condition for the function passing through the determining yield point was set. In Table 3, function coefficients (1) for S235JR + AR steel grade are placed.(1)$$\sigma =K\cdot \varepsilon^{m_{1}}\cdot \exp^{\frac{m_{2}}{\varepsilon }}\cdot \exp^{\varepsilon \cdot m_{3}}$$

Here: *σ*—stress; *ε*—true strain; *K*, *m*_1_*–m*_3_—coefficient of function. 

**Table 3 materials-17-00052-t003:** Coefficients of function (1).

*K* [MPa]	*m* _1_	*m* _2_	*m* _3_	Δmax (MPa)	Δave (MPa)	δave (%)
432.38	0.106616532	0	0.096715784	34.2	16.2	4.49

The second stage of this research work was concerned with measuring the sheet metal geometry after unwinding from the coil using Creaform’s MetraSCAN 3D (Lévis, QC, Canada) optical laser scanner, operating based on laser technology that works together with C-Track device to allow the scanned components to be precisely measured. In combination with the HandyPROBE measuring device, it is possible to scan the sheet metal within a length range of up to 10 m. The accuracy of scanning performed was 0.05 mm. Within the research framework, scanning of the sheet metal reference portion with a length of 3 m was performed. Obtaining the sheet metal batch’s real shape enabled geometry defects occurring to be defined and then the development of the sheet metal computer models based on the 3D sheet metal surface scanning. CAD program was used to process resulting scanned surfaces in the form of *.stl files (objects constructed from triangular meshes), which allowed meshes to be converted into the poly-surfaces described by mathematical equations. The aim of the research carried out was to develop computer models for the numerical modeling of the straightening process in the roll pre-leveler. 

The computer simulation of the leveling process was carried out with the use of an elasto-plastic model in the triaxial state of strain by using the ForgeNxT^®^ v2.1 Transvalor program, whereas the properties of the deformed material were described according to the Norton–Hoff [32,33] conservation law. The application of the computer program ForgeNxT^®^ using the thermo-mechanical models that it contains requires the definition of boundary conditions, which are decisive to the correctness of numerical computation. The theoretical analysis was performed for the following conditions: friction coefficient, *μ*—0.15 (according to the ForgeNxT^®^ v2.1 Transvalor database and [34]); sheet temperature—20 °C; Poisson coefficient—0.3, leveling velocity 45 m/min., number of elements 210,000; specific heat—480 J/(kg·K); density—7850 kg/m^3^; conductivity—29.9 W/(m·K).

## 3. Sheet Geometry 3D Scanning Results

Within the research framework, detailed measurements of the sheet metal geometry (3D scanning) were performed for the selected dimensions and steel grades. This enabled the real size of geometrical defects in the sheet metal after unwinding from the coil to be defined. For the selected sheet metal portions, scanning was performed in the middle and end portions of the coil characterized by the smallest and greatest amplitudes of cross and edge waviness. Based on the performed sheet metal surface scanning with a portion length not less than 3 m, mapped computer surfaces constructed from triangular meshes were obtained. As the sheet metal at the end of the coil was characterized by the greatest defects, the research results presented herein refer to this coil portion.

In Figure 2, the sheet metal to be scanned after unwinding from the coil is shown. The defect analysis carried out showed that the dominating defects of the sheet metal after unwinding from the coil were longitudinal waviness and edge waviness. The areas were observed wherein flexure was in an upward (negative) direction, which is particularly dangerous in the straightening process. The research performed on sheet metal geometry allowed for the conclusion that only defects in central waviness and cross waviness and flexure spots caused by unwinder tensioning roll interaction were present.

In Figure 3a, the exemplary scanned surfaces from the coil end (areas with the greatest geometry defects) are shown. On the scanned surfaces from the coil end, different types of defects of manufacturing origin, but also a result of the sheet metal unwinding from the coil, are clearly visible. A listing of 3D measurement results obtained by scanning is shown in Table 4.

## 4. Pre-Leveler Geometric Model Development

Within the framework of the next research stage, computational models of the sheet metal were developed based on the obtained scanned surfaces. The scanned surfaces utilized in triangular mesh format are of little usefulness due to surface mapping’s high density. Because of the fact that the computation models of the sheet metal are intended for numerical modeling, triangular mesh density must be lower. Therefore, the triangle meshes obtained were transformed in the CAD program into the poly-surfaces described by mathematical equations. Then, mathematical surfaces were cut to the sheet metal format of 3 × 1500 × 3000 mm. The computation model of the sheet metal from the coil end is shown in Figure 3b.

Research conducted concerning the sheet metal geometry after unwinding from the coil allowed defects present in the sheet metal to be identified. Three-dimensional scanning utilization for the sheet metal’s real geometry mapping allowed the sheet metal computation model, which was used to analyze the straightening process in the roll pre-leveler with the use of an FEM-based computer program, to be developed.

The straight sheet metal obtained after the roll pre-leveler’s use requires correct positioning of the flexure rolls. In the available literature, there are no relationships wherein it is possible to design flexure roll positioning. Therefore, within the research framework, the mathematical model was developed based on when it was possible to determine the positioning of the straightening rolls. In Figure 4, the geometric model for the five-roller pre-leveler which was the basis for the development of the mathematical model is presented. The model was developed using IRONCAD 2018 software. The scheme of the roll pre-leveler that was used to develop the mathematical model is shown in Figure 5.

The developed model, described by Equations (2) and (3), is based on engineering relationships allowing approximate real strain and average stress at the sheet metal flexure spots in the roll leveler to be calculated:(2)$$\varepsilon =\sum_{i=1}^{n}\ln\left[\frac{2\cdot \arcsin\left(\frac{0.5\cdot L_{i}}{R_{i}+0.5\cdot G}\right)/\left(R_{i}+0.5\cdot G\right)}{2\cdot \arcsin\left(\frac{0.5\cdot L_{i}}{R_{i}-0.5\cdot G}\right)/\left(R_{i}-0.5\cdot G\right)}\right]\cdot L_{i}\cdot \left(\frac{1}{G}+\frac{1}{n}\right)$$
(3)$$\sigma_{i}=\varepsilon_{i}\cdot L_{i}\cdot \left(\frac{1}{G}+\frac{1}{n}\right)\cdot \sigma (\varepsilon )$$where: *L_i_*—subsequent arc lengths obtained on the base of CAD design; *R_i_*—arc radii obtained on the base of CAD design; *G*—sheet metal thickness; *n*—number of arcs; *σ*(*ε*)—yielding curve function and Young’s modulus.

To determine strain and stress, it is necessary to know the curve lengths and the sheet metal flexure radius values between the rolls, as well as the sheet metal thickness. Based on the length difference between the lower and upper edges of the sheet metal (natural logarithm), it is possible to calculate real strain. Such information was obtained from the theoretical CAD model shown in Figure 5. There exist four arcs and radii for the five-roller leveler. It was assumed for the model that only the arc length half is subjected to yielding strain. The sum of all strains approximately corresponds to the effective strain being obtained in the roll levelers. A mathematical model based on function (1) and coefficients developed by approximation was utilized to determine the stress value in flexure zones. It is necessary to know the Young’s modulus value in the model. The model’s limitation is the assumption of a constant yielding strain zone of 50%, due to the fact that during the metal sheet flexure, the yielding strain zone is on average 50% of the flexure arc length.

The sheet metal end was used as a reference geometric model. In Figure 6, the sheet metal’s real longitudinal profile obtained by 3D scanning is presented. 

It follows from the data shown in Figure 6 that for the range from 1650 mm to 3000 mm of the reference portion length, the longitudinal waviness for the center and sides coincides, whereas, for lengths between 500 mm and 1500 mm, greater differences in the nature of the longitudinal waviness were found. This is a typical hot-rolled sheet metal profile after being unwound from the coil. Based on the analysis of longitudinal geometry changes, an averaged model of the sheet metal was developed. The model was used as input for the straightening process numerical simulations. The roll arrangement for different variants is shown in Table 5.

## 5. Results and Discussion

Utilizing relations (2) and (3) and the assumed straightening variants, the sheet metal straightening numerical simulations were performed. In Figure 7, the results of the sheet metal of S235JR + AR grade straightening numerical simulations in the roll pre-leveler were presented, a schematic diagram of which is shown in Figure 4 and Figure 5. Utilizing a mathematical model of the roll arrangement, straightening process numerical simulations were performed for four variants. In Table 6, Table 7, Table 8 and Table 9, calculation results are placed. Bending radii *R_i_* and bend deflection *t_i_* of the straightened sheet metal were determined in tables and, based on these parameters, strains and stresses during sheet metal flexure after the *i*-th roll were determined. Based on component strains, the total strain in the sheet metal after straightening was determined. Roll arrangement was selected in such a manner that real strain exceeded the yielding strength value to ensure that the sheet metal was straightened. Furthermore, it is a prerequisite that necessary strain is minimized so that the sheet metal is not strengthened excessively. It is also a prerequisite for the obtained sheet metal to be straightened geometrically as much as possible.

In order to fully verify the mathematical model of the straightening rolls, four variants of numerical simulations of the sheet metal straightening were performed. In no case was full straightening of the reference sheet metal portion of S235JR + AR steel achieved. It turned out that variant 3 was the most advantageous in terms of sheet metal straightness. In Figure 7, the calculation results for variant 1 are shown. It follows from the figure analyzed that the longitudinal waviness along the entire length of the reference sheet metal was homogenized. However, the sheet metal flatness was not achieved.

In Figure 8, the sheet metal’s average profile after roll straightening and the trend obtained as a result of curve approximation were compared.

It follows from the data shown in Figure 8 that there are local disturbances (deviations) of the sheet metal profile as compared to the average sheet metal curvature. The subsequent sheet metal profile correction is presented in Figure 9. Utilizing a developed mathematical model, roll arrangement correction in the roll straightener was introduced. In Figure 9, the results of the sheet metal shape calculation obtained for variant 3 are presented.

It follows from Figure 9 that the sheet metal shape along the prevailing length thereof was homogenized. Simultaneously, on the drawing analyzed, there are areas characterized by the sheet metal profile deviation. The results obtained confirm the necessity for using the additional roll straightener or tension straightening in order to achieve the desired sheet metal flatness.

Straightening efficiency resulting from preliminary calculations should be 0.04–0.1 of real strain. In the case of variant 4, the excessive roll pressure results in the straightened sheet metal flexure (deforming) at the strain of 0.3 and stresses significantly exceeding the steel yield strength.

Numerical modeling results showed that the roll arrangement according to variants 1 and 2 did not produce satisfactory results. The sheet metal flexures set during straightening were too small, which caused some areas of the sheet metal to not yield. This was due to the sheet metal’s excessive initial waviness. Waviness pitch corresponding to roll spacing is not conducive to effective straightening, and these areas did not yield completely, which is shown in Figure 10 and Figure 11.

Roll arrangement applying (variant 3) straightening to the sheet metal with the greatest waviness (from the coil end) provided the best results (Figure 12). However, the sheet metal obtained was characterized by flexure, and the local deviations from straightness were ±3 mm, which is shown in Figure 8 and Figure 12, whereas the high roll pressures applied for variant 4, as an example, result in sheet metal strain but also introduce high stresses between the feeding and straightening rolls, which may cause uncontrollable changes in the sheet metal geometry (Figure 13). Straightening efficiency according to variant 4 is low, and metal sheet strengthening is the greatest, which may be evidenced by high strain values.

## 6. Conclusions

The research carried out relative to the sheet metal geometry after being unwound from the coil allowed defects occurring in the sheets to be identified. Three-dimensional scanning utilized for the real geometry mapping of the sheet metal allowed the sheet metal computation models to be developed, which were used to analyze the straightening process in the roll pre-leveler with the use of an FEM-based computer program.

The mathematical model developed, combined with the FEM simulation results, ensured the introduction of individual roll arrangement corrections. This enabled the roll arrangement selection and elaboration of straightening schemes that included parameters for roll arrangement.

The research carried out allowed straightening roll arrangement and stress and strain determination during straightening to be precisely selected. It should be noted that the model developed relates to the straight initial sheet metal; in case of waviness occurring in the sheet metal batch during yielding (real conditions), straightening may be less effective, and the sheet metal yielding may not occur along the entire length. This is related to the sheet metal profile, the number and spacing of the rolls, and the diameter thereof.

## Figures and Tables

**Figure 1 materials-17-00052-f001:**
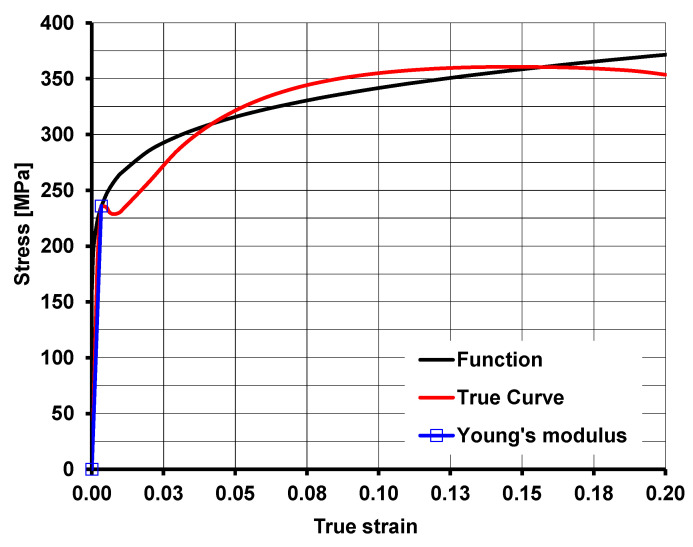
Graphic presentation of the S235JR + AR steel grade yielding stress mathematical model (red—real curve; blue—Young’s modulus; black—four-parameter yielding stress function).

**Figure 2 materials-17-00052-f002:**
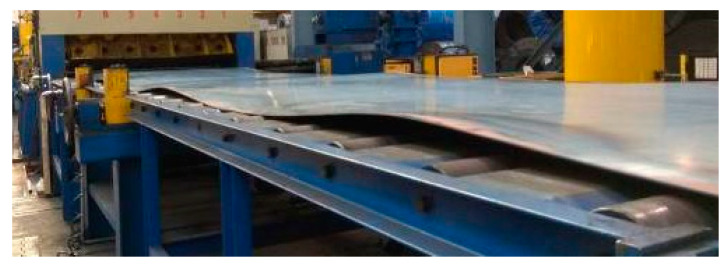
The sheet metal edge and longitudinal waviness examples after unwinding from the coil of S235JR + AR steel grade with dimensions of 3 × 1500 mm.

**Figure 3 materials-17-00052-f003:**
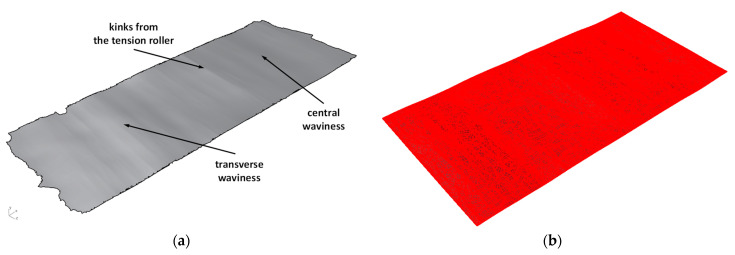
The sheet metal surface (**a**) from the coil end obtained by 3D scanning and (**b**) after numerical processing of edge and end cutting transformed into the poly-surface.

**Figure 4 materials-17-00052-f004:**
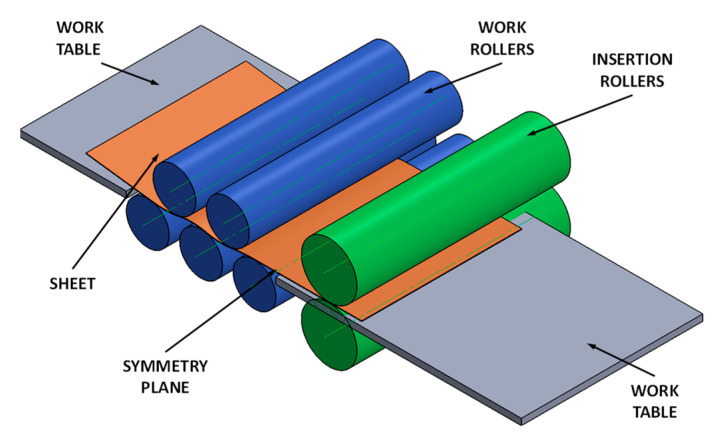
The view of 5-roller pre-leveler and geometric model schemes used for the sheet metal straightening computation simulation.

**Figure 5 materials-17-00052-f005:**
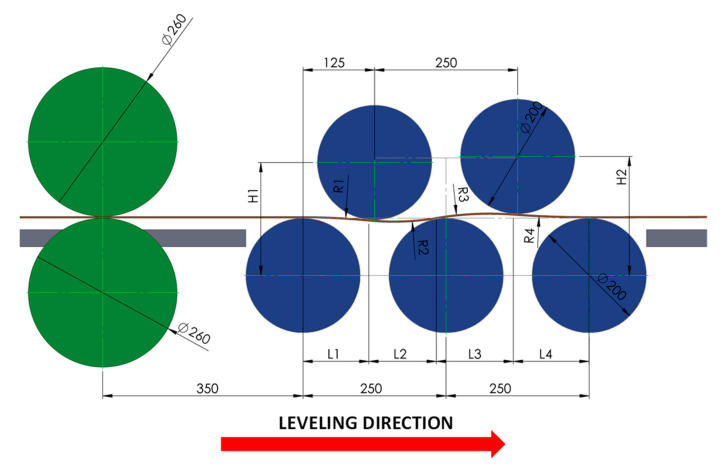
Dimensions and position of the rollers in the 5-roller pre-leveler.

**Figure 6 materials-17-00052-f006:**
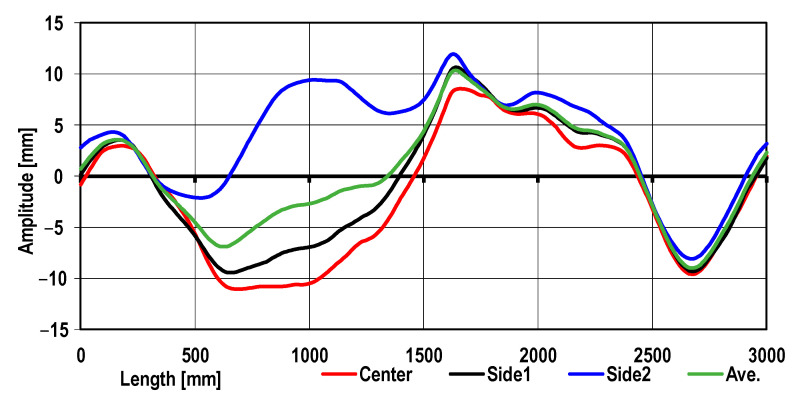
The sheet metal’s longitudinal profile after unwinding from the coil obtained by 3D scanning.

**Figure 7 materials-17-00052-f007:**
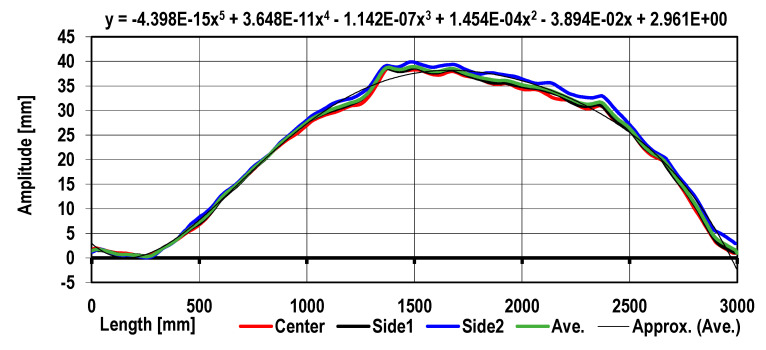
The sheet metal profile after straightening in the 5-roller levelers obtained as a result of computation simulation (flexure approximation by a polynomial function of degree 5).

**Figure 8 materials-17-00052-f008:**
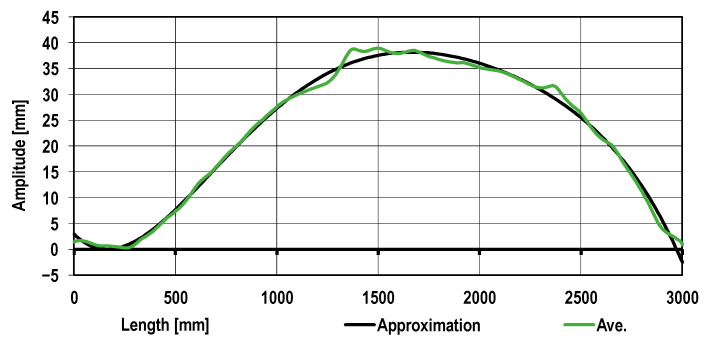
The sheet metal average longitudinal profile compared to the approximated curve obtained.

**Figure 9 materials-17-00052-f009:**
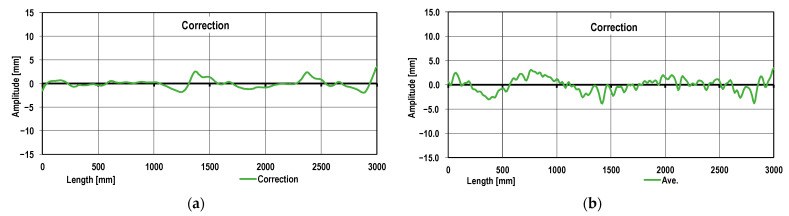
The sheet metal profile local deviation analysis taking into account the sheet metal approximated curvature: (**a**) the numerical modeling; (**b**) the real sheet metal profile.

**Figure 10 materials-17-00052-f010:**
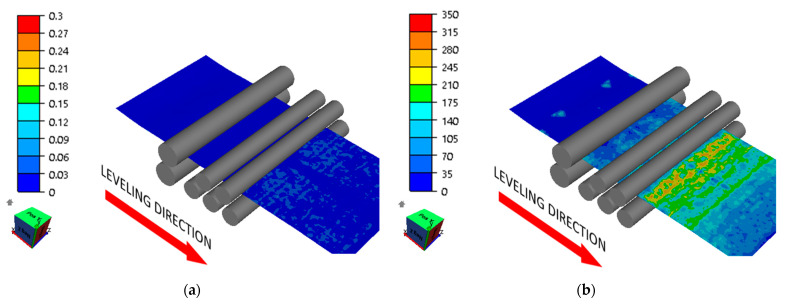
Distribution of (**a**) effective strain (-) and (**b**) effective stress (MPa) in the sheet being straightened (variant 1).

**Figure 11 materials-17-00052-f011:**
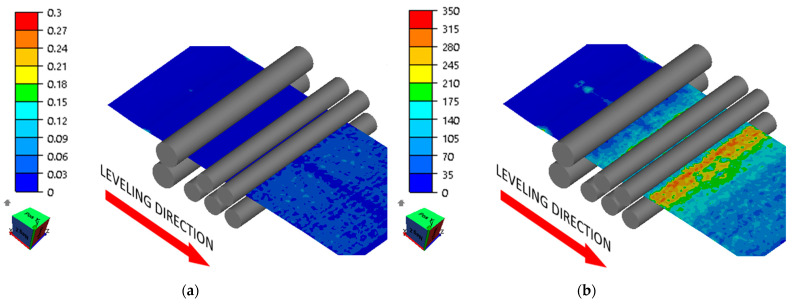
Distribution of (**a**) effective strain (-) and (**b**) effective stress (MPa) in the sheet being straightened (variant 2).

**Figure 12 materials-17-00052-f012:**
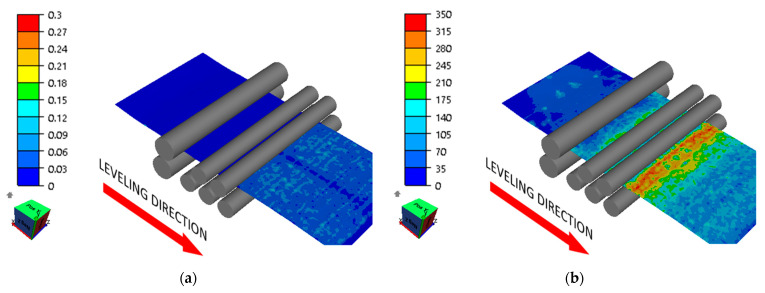
Distribution of (**a**) effective strain (-) and (**b**) effective stress (MPa) in the sheet being straightened (variant 3).

**Figure 13 materials-17-00052-f013:**
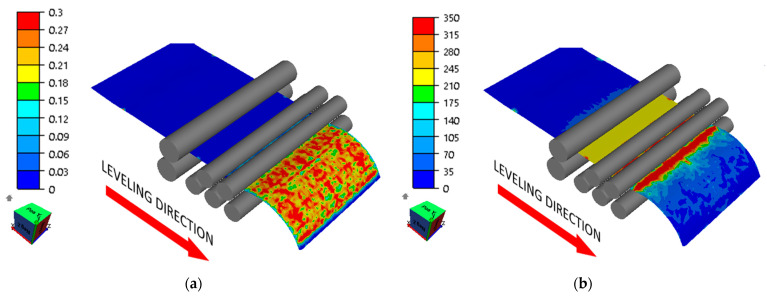
Distribution of (**a**) effective strain (-) and (**b**) effective stress (MPa) in the sheet being straightened (variant 4).

**Table 1 materials-17-00052-t001:** Chemical composition of the steel used for the tests.

Material	Chemical Composition, % Mass.					
**S235JR + AR**	**Fe**	**C**	**Mn**	**Si**	**P**	**S**
	ball.	0.22	1.62	0.05	0.052	0.048

**Table 2 materials-17-00052-t002:** Mechanical properties of the steel grade tested.

Steel Grade	Yield Stress Y_0.2_ (MPa)	Ultimate Tensile Stress UTS (Mpa)	Young’s Modulus E (GPa)	Max. Absolute Error Δ (MPa)	Max. Relative Error δ (%)
S235JR + AR	235	360	181	±8.25	2.29

**Table 4 materials-17-00052-t004:** Geometrical parameters of the sheet metal waviness measurement.

Sheet Thickness (mm)	Sheet Width (mm)	Average of Height of Waviness (mm)		Period of Waviness Occurrence l (mm)
		Middle of the Coil	End of the Coil	
3	1500	15.2	21.5	irregular

**Table 5 materials-17-00052-t005:** Roll arrangement variants used for metal sheet straightening in the 5-roller pre-leveler.

Variant	H1 (mm)	H2 (mm)	Rollers Diameter (mm)	Distance between Rollers (mm)	Diameter of Insertion Rollers (mm)
1	200.5	202.5	200	250	250
2	200.0	202.0	200	250	250
3	199.0	202.0	200	250	250
4	194.0	197.0	200	250	250

**Table 6 materials-17-00052-t006:** Strain and stress in the sheet metal flexure spots for variant 1.

*i*-th Roller	Roller Distance (mm)	Roller Diameter (mm)	Arc Length *L_i_* (mm)	Bending Radius *R_i_* (mm)	Deflection Arrow *t_i_* (mm)	Young’s Modulus (GPa)	Ye (MPa)	Thickness G (mm)	Strain	Stress (MPa)
1	250	200	120.9	3024.7	0.6	181	235	3	0.0037	232.3
2	250	200	121.4	1055.8	1.7	181	235	3	0.0108	239.1
3	250	200	126.6	938.8	2.1	181	235	3	0.0126	243.5
4	250	200	91.9	1380.6	0.8	181	235	3	0.0062	229.9
								Total	0.0334	

**Table 7 materials-17-00052-t007:** Strain and stress in the sheet metal flexure spots for variant 2.

*i*-th Roller	Roller Distance (mm)	Roller Diameter (mm)	Arc Length *L_i_* (mm)	Bending Radius *R_i_* (mm)	Deflection Arrow *t_i_* (mm)	Young’s Modulus (GPa)	Ye (MPa)	Thickness G (mm)	Strain	Stress (MPa)
1	250	200	120.1	2504.2	0.7	181	235	3	0.0045	229.9
2	250	200	120.8	882.7	2.1	181	235	3	0.0128	244.0
3	250	200	126.2	757.1	2.6	181	235	3	0.0156	250.5
4	250	200	97.9	959.1	1.2	181	235	3	0.0096	236.2
								Total	0.0425	

**Table 8 materials-17-00052-t008:** Strain and stress in the sheet metal flexure spots for variant 3.

*i*-th Roller	Roller Distance (mm)	Roller Diameter (mm)	Arc Length *L_i_* (mm)	Bending Radius *R_i_* (mm)	Deflection Arrow *t_i_* (mm)	Young’s Modulus (GPa)	Ye (MPa)	Thickness G (mm)	Strain	Stress (MPa)
1	250	200	118.5	1853.6	0.9	181	235	3	0.0060	229.7
2	250	200	119.7	666.7	2.7	181	235	3	0.0168	253.3
3	250	200	126.9	595.5	3.4	181	235	3	0.0200	260.3
4	250	200	101.8	760.1	1.7	181	235	3	0.0126	243.3
								Total	0.0554	

**Table 9 materials-17-00052-t009:** Strain and stress in the sheet metal flexure spots for variant 4.

*i*-th Roller	Roller Distance (mm)	Roller Diameter (mm)	Arc Length *L_i_* (mm)	Bending Radius *R_i_* (mm)	Deflection Arrow *t_i_* (mm)	Young’s Modulus (GPa)	Ye (MPa)	Thickness G (mm)	Strain	Stress (MPa)
1	250	200	110.5	771.1	2.0	181	235	3	0.0134	245.4
2	250	200	116.1	310.0	5.4	181	235	3	0.0351	289.7
3	250	200	128.1	294.8	7.0	181	235	3	0.0407	299.4
4	250	200	111.7	335.6	4.6	181	235	3	0.0312	282.6
								Total	0.1205	

## Data Availability

Data are contained within the article.

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
