# Peer review of "The Selection of Leveler Parameters Using FEM Simulation"

_materials, 2023, doi:10.3390/ma17010052_

Round 1

Reviewer 1 Report

Comments and Suggestions for Authors

General review:

Overall, the authors tried to select parameters for the roll pre-leveler to provide sheet metal waviness reduction after unwinding from the coil. Straightening parameters were selected based on the results of numerical simulations with using of an FEM-based computer program. The material used for research was the hot-rolled sheet metal of grade S235JR+AR with a thickness of 3 mm and width of 1500 mm after unwinding from the coil. The mathematical model was developed to determine straightening roll arrangements in the pre-leveler. It enabled roll arrangement selection and straightening scheme to be elaborated. The innovative feature was to conduct straightening numerical simulations for the real sheet metal geometric models obtained as a result of a 3D laser scanning which ensured the accuracy of numerical calculations to be increased. However, before it can be published, I have some questions about this article and some suggestions:

Minor review:

1. Several sentences starting in Introduction were too too general to distinguish this paper. Please start directly to mention the main issues in this paper.

2. You should more mention the experimental procedures and how the S235JR+AR are made and heat treated in the process in more detail.

3. Some of the references are too old and thus replace them.

Otherwise, the author has addressed well most of the explanations. Overall, this is a good study. I supposed it can be considered for publication in Materials after minor revision.

Author Response

  1. Several sentences starting in Introduction were too general to distinguish this paper. Please start directly to mention the main issues in this paper.

The introduction was changed and additional literature was added

  1. You should more mention the experimental procedures and how the S235JR+AR are made and heat treated in the process in more detail.

Hot-rolled sheets of the S235JR grade can be annealed after the rolling process, however, due to the reduction in production costs, they are rolled into coils without additional heat treatment. These sheets cool in a cold store at ambient temperature. During hot rolling, the rolls are cooled with water and most often the edges of the sheet are cooled faster than the central part of the sheet, which may cause different properties in the central and side parts. During rolling, the deformation state at the edges is different than in the center of the sheet, hence differences in the properties of steel may occur.

The samples for mechanical property testing were taken from the sheets from the be-ginning, middle, and end of the coil in the middle sheet metal portion and near the sheet metal side edges. Three samples were prepared from each location. Due to the significant number of test samples, the testing methodology of two samples per area was assumed. In case of repeating results, the test series consisted of 2 samples. If a difference in tensile force values of more than 3% was obtained, a series including additional tests were used with the use of the third sample.

Research work second stage concerning the sheet metal geometry measurement after unwinding from the coil using Creaform's MetraSCAN 3D (Canada) optical-laser scanner operating based on laser technology which working together with C-Track device allows the scanned components to be precisely measured. In combination with the HandyPROBE measuring device, it is possible to scan the sheet metal within a length range of up to 10 m. The accuracy of scanning performed was 0.05 mm. Within the research framework, scanning of the sheet metal reference portion with a length of 3 m was performed. Ob-taining the batch sheet metal real shape enabled geometry defects occurring to be defined and then the development of the sheet metal computer models based on the 3D sheet met-al surface scanning. CAD program was used to process resulting scanned surfaces in the form of *.stl files (objects constructed from triangular meshes) which allowed meshes to be converted into the poly-surfaces described by mathematical equations.

  1. Some of the references are too old and thus replace them.

The latest literature was added

Reviewer 2 Report

Comments and Suggestions for Authors

The objective of the study was to identify optimal parameters for the roll pre-leveler, aiming to reduce waviness in sheet metal after unwinding from the coil. Straightening parameters were chosen based on the outcomes of numerical simulations using a FEM analysis. The research utilized hot-rolled sheet metal of grade S235JR+AR after unwinding from the coil. A mathematical model was devised to determine the arrangement of straightening rolls in the pre-leveler, allowing for the selection of roll arrangements and the development of a straightening scheme. An aspect of the study involved performing numerical simulations for real sheet metal geometric models obtained through 3D laser scanning, thereby enhancing the accuracy of the numerical calculations.

Comments:

-          Introduction must be significantly improved. A description of current research in the field of numerical simulations is missing.

-          According to which standard was the static tensile test performed?

-          Table 2, Yield stress – use standard decimal point 0.2 instead of 0,2.

-          Was the coefficient of friction value (0.15) on page 4, line 138 measured or chosen from other study?

-          Figures 10-13, legends are hard to read, must be improved.

Author Response

- Introduction must be significantly improved. A description of current research in the field of numerical simulations is missing.

Introduction was changed. An additional literature was added

- According to which standard was the static tensile test performed?

The latest standard was added

PN-EN ISO 6892-1:2020-05, Metale -- Próba rozciÄ…gania -- Część 1: Metoda badania w temperaturze pokojowej, (in Polish).

- Table 2, Yield stress – use standard decimal point 0.2 instead of 0,2.

It was corrected

- Was the coefficient of friction value (0.15) on page 4, line 138 measured or chosen from other study?

The friction coefficient was adopted from the material database of the ForgeNxT® program. For cold processes without lubrication, the friction coefficient is in the range of 0.1 - 0.2 [1] depending on the roughness of the contact surface. For roughness Ra = 0.63 for steel rollers, the assumed coefficient seems to be appropriate for the assumed actual deformation.

[1] Trzepieciński, T. A Study of the Coefficient of Friction in Steel Sheets Forming. Metals 2019, 9, 988.

- Figures 10-13, legends are hard to read, must be improved.

Figures were modified

Round 2

Reviewer 2 Report

Comments and Suggestions for Authors

The authors improved their manuscript, I have no further comments. I recommend the manuscript for publication.